
# A quantitative assessment of air-sea heat flux trends from ERA5 since 1950 in the North Atlantic basin

Johannes Mayer[1], Leopold Haimberger[1], and Michael Mayer[1,2]

[1]Department of Meteorology and Geophysics, University of Vienna, Vienna, Austria
[2]European Centre for Medium-Range Weather Forecasts, Bonn, Germany

**Correspondence:** Johannes Mayer (johannes.mayer@univie.ac.at)

**Abstract.** This work aims to investigate the temporal stability and reliability of trends in air-sea heat fluxes from ERA5 forecasts over the North Atlantic basin for the period 1950–2019. Driving forces of the trends are investigated using analyzed state quantities from ERA5. Estimating trends from reanalysis data can be challenging as changes in the observing system may introduce temporal inconsistencies. To this end, the impact of analysis increments is discussed. For individual sub-regions in the North Atlantic basin, parametrization formulas for latent and sensible heat fluxes are linearized to quantitatively attribute trends to long-term changes in wind speed, moisture, and temperature. Our results suggest good temporal stability and reliability of air-sea heat fluxes from ERA5 forecasts on sub-basin scale and below. Regional averages show that trends are largely driven by changes in the skin temperature and atmospheric advection (e.g., of warmer or drier air masses). The influence of climate variability modes, such as the North Atlantic Oscillation (NAO) and Atlantic Multidecadal Oscillation on the found patterns is discussed as well. Results indicate a significant impact on trends in the Irminger and Labrador Sea associated with more positive NAO phases during the past 4 decades. Finally, we use basin-wide trends of air-sea heat fluxes in combination with an observational ocean heat content estimate to provide an energy-budget-based trend estimate of the Atlantic meridional overturning circulation (AMOC). A decrease of area-averaged air-sea heat fluxes in the North Atlantic basin suggests a decline of the AMOC over the study period. However, basin-wide flux trends are deemed partially artificial, as indicated by temporally varying moisture increments. Thus, the exact magnitude of change is uncertain, but its sign appears robust and adds complementary evidence that the AMOC has weakened over the past 70 years.

## 1 Introduction

The North Atlantic Ocean plays a central role for weather and climate in Europe and eastern North America. For instance, the formation of tropical cyclones and severe weather systems along the Gulf Stream and its extension have a significant impact on our economy, agriculture, and society. The North Atlantic Oscillation [NAO; Hurrell (1995); Visbeck et al. (2001)], a periodic change in strength of Azores High and Icelandic Low, impacts the moisture transport in the northern hemisphere on seasonal timescales and thus influences temperature and precipitation in wide areas of Europe and North America. In addition, the Gulf Stream current in the western North Atlantic is responsible for the poleward transport of oceanic energy that is taken up in tropical latitudes and released to the atmosphere further north via air-sea fluxes. The associated cooling of the waters is



required to trigger deep water formation, which is the main driver of the Atlantic meridional overturning circulation [AMOC; Rahmstorf et al. (2015)] and consequently also of the global thermohaline circulation. Previous research used climate models to demonstrate that the AMOC has declined over past decades as a result of anthropogenic global warming, which could have further effects on storms and weather patterns (Fox-Kemper et al., 2021). However, direct observations of the AMOC are limited in time and do not show clear evidence of an externally forced slowdown (Baehr et al., 2007; Roberts et al., 2014;

Worthington et al., 2021).

In all these processes, the exchange of energy (and momentum) between atmosphere and underlying ocean is of vital importance. Long-term changes in air-sea heat fluxes over the North Atlantic Ocean can thus have a wide range of implications. Consequently, it is of high relevance to accurately estimate air-sea heat flux trends, which also helps to understand numerous aspects of climate variability in the North Atlantic basin. Nonetheless, observation-based estimates are spa-

tially limited and are available only for the past two to three decades (see, e.g., oceansites flux data provided by NOAA, https://www.pmel.noaa.gov/gtmba/) making the distinction between anthropogenic changes and natural variability on decadal to multi-decadal timescales demanding. For instance, the Atlantic multidecadal oscillation [AMO; Kerr (2000)] describes a natural variability mode of basin-wide SST anomalies on timescales of 70–80 years so that its long-term effect on air-sea heat fluxes can not be determined adequately from observations.

An alternative to observations is given by recent reanalysis data such as the fifth generation global reanalysis data produced by ECMWF [ERA5; Hersbach et al. (2020)] providing global gridded data for more than seven decades thanks to its recent back-extension (Bell et al., 2021). Reanalysis data are constructed from past model forecasts constrained by observational data (such as in-situ, satellite, airplanes, or radiosondes) through data assimilation, which warrants optimal combination and reduction of biases. However, changes in the observing system can result in temporal discontinuities and introduce increments

between forecasts and analyses that may alter the atmospheric state [moisture, temperature, and wind; see Chiodo and Haimberger (2010) and Mayer et al. (2021)] and consequently also air-sea heat fluxes making climate trend studies with reanalysis data challenging.

Reanalysis data can also be used to indirectly estimate air-sea heat fluxes by evaluating the atmospheric energy budget (Trenberth, 1991; Mayer et al., 2016; Trenberth and Fasullo, 2017; Liu et al., 2017; Mayer et al., 2019, 2021, 2022; Liu

et al., 2022). This method does not inherently reduce temporal discontinuities, but allows the application of a global wind field correction (Trenberth, 1991; Fasullo and Trenberth, 2008; Mayer and Haimberger, 2012), which diminishes both artificial noise over high topography and temporal discontinuities introduced by changes in the observing system (Mayer et al., 2021). Although air-sea heat fluxes derived from ERA5 in this way are available only for the period from 1985 onward, they are proven to be temporally relatively stable over the global ocean (around $1.7 \, \mathrm{W \, m^{-2}}$ mean based on 1985–2018) and thus can

serve as reference to test the reliability of other commonly used air-sea heat flux products (Mayer et al., 2021, 2022).

This work aims to investigate the reliability and temporal stability of long-term trends of winter months (December–February) net surface heat fluxes based on ERA5 data over the North Atlantic ocean during 1950–2019. Main drivers of trends in latent and sensible heat fluxes are identified based on analyzed state quantities, and the impact of the assimilation process and climate variability modes, such as NAO and AMO, are discussed. Whenever possible, net air-sea heat fluxes from





ERA5 are compared with indirect estimates from Mayer et al. (2022). In four individual sub-regions, turbulent air-sea heat flux trends are quantitatively attributed to long-term changes in wind speed, moisture, and temperature using linearized flux parameterization formulae. Finally, we use basin-wide air-sea heat fluxes from ERA5 forecasts and an observation-based ocean product to indirectly estimate trends of the AMOC over the past 70 years, and discuss sources of uncertainties and reliability of the trend estimate.

The data we use in this study are introduced in section 2. Section 3 describes the methodology. Results are presented in section 4 and summarized and discussed in section 5.

## 2   Data

The data we primarily use in this study are from ECMWF's most recent reanalysis ERA5 (Hersbach et al., 2020). ERA5 provides a variety of meteorological variables as 12-hourly twice-daily forecasts as well as analyzed state quantities on a

Gaussian grid equivalent to 0.25 degree spatial resolution [see Hersbach et al. (2020) for details]. Individual components of the net air-sea heat flux (i.e., short-wave and long-wave radiation, and sensible and latent heat flux) are taken as monthly means and are available only as forecasts (denoted as model-based fluxes from ERA5 forecasts). We also use single-level atmospheric moisture, temperature, pressure, and 10 metre wind fields as monthly means from both analyses and forecasts to compute surface heat fluxes using parameterizations as implemented in the Integrated Forecast System [IFS; see ECMWF (2021)],

which allows to estimate the role of changes in single input parameters. The 3D horizontal wind fields used to compute the meridional mass stream function are also taken from ERA5 but on pressure levels and regular 0.25×0.25 grid.

Whenever possible, we compare model-based fluxes with indirectly estimated net surface heat fluxes (denoted as inferred surface fluxes) from Mayer et al. (2022), which are derived from the DEEP-C TOA flux product [see Liu et al. (2020)] and atmospheric energy transports from Mayer et al. (2022b) [see also Mayer et al. (2021) and section 2 in Mayer et al. (2022) for

details of the computation and assessment]. Inferred surface fluxes are provided as monthly averages on a one degree regular grid covering 1985–2020.

Observationally constrained ocean heat content (OHC) data are provided by the Institute of Atmospheric Physics [IAP; Cheng et al. (2017); available at http://www.ocean.iap.ac.cn/ftp/cheng/IAP_Ocean_heat_content_0_2000m/] on a regular 1×1° grid covering the whole study period 1950–2019. We use the 0–300m OHC monthly data for the correlation and comparison

with the skin temperature from ERA5, and the 0–2000m OHC to indirectly estimate the AMOC trend from the oceanic heat budget. The IAP dataset is constructed based on a modified version of the ensemble optimal interpolation method proposed by Cheng and Zhu (2016). Sampling errors are minimized using observational data and a prior guess from Coupled Model Intercomparison Project Phase 5 (CMIP5) multimodel simulations. The small sampling error indicates a robust reconstruction of the temperature signal in all ocean basins. Furthermore, the dataset is bias-corrected using in situ observations (CBT and

MBT data) and thus appears well suited for our evaluations.

In addition, we use mooring-derived and volume-conserving monthly ocean heat transport (OHT) estimates from Tsubouchi et al. (2018) and Tsubouchi et al. (2020) as northern choke point of the oceanic heat budget. Tsubouchi et al. (2018) offer trans-





port estimates from the Davis Strait (DS; see http://metadata.nmdc.no/metadata-api/landingpage/0a2ae0e42ef7af767a920811e83784b1)
covering the period 1993–2016. Tsubouchi et al. (2020) provide observations for 2005–2010 from the Fram Strait and Barents

Sea Opening (FS and BSO; see https://doi.pangaea.de/10.1594/PANGAEA.909966).

## 3 Methods

### 3.1 Bulk formulas

In this study, sensible and latent heat fluxes are taken from 12-hourly ERA5 forecasts. However, to i) estimate the relationship
between analysis increments and long-term flux trends, and ii) for regression on input variables, we compute fluxes from

scratch as described in the following.

Net surface energy fluxes are the sum of radiative and turbulent heat fluxes. Radiative fluxes contain short- and long-wave
radiation and are not discussed in detail here. Turbulent heat fluxes are the sum of latent and sensible heat fluxes and can be
approximated by the commonly used bulk formulas [see Fairall et al. (2003), Cronin et al. (2019), and ECMWF (2021)], which
are written as follows


$$F_{LH} = C_Q \, \rho \, |U_{ml}| \, (L_v \, q_{ml} - L_v \, q_{sfc})$$

$$\text{with} \quad F_{LH} = F_{LH}(\rho, U_{10m}, q_{ml}, q_{sfc}, t), \ q_{sfc} = q_{sfc}(p_{sfc}, T_{skin}) \quad (1)$$

$$F_{SH} = C_H \, \rho \, |U_{ml}| \, (c_p \, T_{ml} - c_p \, T_{skin} + g \, z_{ml}) \quad \text{with} \quad F_{SH} = F_{SH}(\rho, U_{10m}, T_{ml}, T_{skin}, t) \quad (2)$$

where $C_Q$ and $C_H$ are non-constant transfer coefficients [see also ECMWF (2021)], $\rho$ is the air density above ocean, $L_v$
($2.5008 \times 10^6$ J kg$^{-1}$) is the latent heat of vaporization, $c_p$ (1004.709 J kg$^{-1}$ K$^{-1}$) is the specific heat capacity of dry air, g
(9.80665 m s$^{-2}$) is the gravitational acceleration, and $z_{ml}$ is the height of the lowest model level. $|U_{ml}|$, $q_{ml}$, and $T_{ml}$ are the
wind speed, specific humidity, and air temperature at lowest model level. $q_{sfc}$ is the surface saturation humidity, and $T_{skin}$ is
the skin temperature (as used in the IFS instead of the sea surface temperature). $q_{sfc}$ depends on surface pressure $p_{sfc}$ and

$T_{skin}$ and can be derived from the Clausius-Clapeyron relation for 100 % relative humidity. Parameters at lowest model level
and above ocean are with good approximation 10 metres above the surface. According to Eq. (1) and (2), fluxes from the
atmosphere into the ocean are positive.

Whenever fluxes are computed with the above formulae, transfer coefficients are indirectly approximated for each grid point
by dividing latent and sensible heat fluxes from ERA5 forecasts with the other terms of the right hand side of Eq. (1) and (2)

(also from forecasts), which has to be done before flux anomalies are computed. This procedure works remarkably well for
the ice-free ocean. Over sea ice, however, differences between model-level and surface quantities can be very small (i.e., mean
climatology of absolute temperature and moisture differences can be $\leq$0.01 K and $\leq$0.01 g kg$^{-1}$) so that the division by small
numbers introduces artificial noise.





Seasonal trends are computed from monthly anomalies by subtracting the climatology of each grid point and subsequently averaging over December–February. The statistical significance of seasonal trends is computed with the 95 % confidence level and consideration of lag-1 autocorrelation. Analysis increments of model-level parameters (temperature and humidity) are calculated from the difference between analyzed state quantities and 12-hourly forecasts valid at the analysis time. The impact of increments on flux trends is then estimated by taking the difference between fluxes computed with analyzed state quantities and with short-term forecasts of those quantities.

In this study, we closely follow the mathematical interpretation of air-sea heat fluxes [i.e., Eq. (1) and (2)] and assume that the derived flux trends are solely caused by changes in their input variables (wind, moisture, and temperature). We are aware that this is not the physically correct interpretation as changes in fluxes also influence the input variables due to their mutual dependency (e.g., an increase of latent heat flux increases the moisture in the lowest model-level, which in turn reduces latent heat fluxes). However, we argue that in equilibrium (i.e., considering long-term changes over multiple decades) this mutual influence is irrelevant for such discussions so that long-term flux trends can be considered as direct result of trends in input variables.

### 3.2 Linearization of turbulent heat fluxes and partial trends

To attribute trends in latent and sensible heat fluxes to trends in their input variables in a more quantitative way, we linearize the bulk formulas by decomposing each input variable x ($\rho$,U,q, and T) into their mean state $\bar{x}$ and deviation from its mean $x'$ [known as Reynolds decomposition; see also Tanimoto et al. (2003) and Yang et al. (2016)]; that is, each input variable on the right side of Eq. (1) and (2) is described by $\bar{x}+x'$. After some calculus, computed turbulent heat fluxes can further be separated into a non-linear and linear part (see appendix A). The latter contains all products with at most one deviation term on which input variables are regressed to obtain a linear relation between trend in surface flux F (sensible or latent heat) and trend of each input variable, which reads as follows

$$\frac{\partial F}{\partial t} = \frac{\partial F}{\partial \rho}\frac{\partial \rho}{\partial t} + \frac{\partial F}{\partial |U_{10m}|}\frac{\partial |U_{10m}|}{\partial t} + \frac{\partial F}{\partial \xi_{ml}}\frac{\partial \xi_{ml}}{\partial t} + \frac{\partial F}{\partial \xi_{sfc}}\frac{\partial \xi_{sfc}}{\partial t} \quad \text{with} \quad F = F(\rho, U_{10m}, \xi_{ml}, \xi_{sfc}, t), \tag{3}$$

where $\xi$ is a placeholder for either q or T. We term the expressions on the right *partial trends*, which are the product of the mean *sensitivity* $\partial F/\partial x$ (i.e., regression of F on x using the whole period of time) and linear trend $\partial x/\partial t$ of input variable x. The mean sensitivity describes how F changes when x is changed. Consequently, partial trends on the right side tell us how much of the flux trend ($\partial F/\partial t$) is explained by the trend in one of the input variables ($\partial x/\partial t$). Note that this procedure neglects trends in transfer coefficients and the non-linear part, which is a sufficient assumption for the purpose of this study as their contribution to the total turbulent heat flux trend is rather small (the linear part explains $\geq$93 % of the total turbulent flux trend). Furthermore, we do not show partial trends of $\rho$ in our evaluations as they are negligibly small.



### 3.3 Study area and box averages

Our general study area includes the ice-free ocean between 0–90° N and 90° W–30° E, but particular focus is laid on four
8×8° boxes for which partial trends of latent and sensible heat fluxes are computed. The four focus regions are located in
the Norwegian Sea (NWS; 68–76° N 2–10° E), over the northern flank of the North Atlantic Warming Hole (NAWH; 55–63°
N 37–29° W), along the Gulf Stream extension (GS, 35–43° N 66–58° W), and in the tropical North Atlantic (TNA; 15–
23° N 40–32° W) as trends in these regions are associated with distinct atmospheric and oceanic thermodynamics (see Fig.
1). Spatially averaging over these areas reduces the variance of fluxes and computational cost while trends are still captured
reasonably well.

### 3.4 Meridional mass stream function

The seasonal mean meridional mass stream function $\Psi_m$ is obtained by

$$\Psi_m = \frac{r_E}{g} \int_{\lambda=0}^{\lambda=2\pi} \int_0^p \bar{v}^\star \, dp \, d\lambda \tag{4}$$

where $r_E$ is the Earth's radius and $\bar{v}^\star$ is the seasonal mean of the deviation of the meridional wind component (on pressure
levels) from its meridional mean, which is vertically integrated from the TOA to the pressure level of interest (p) and over all
longitudes $\lambda$.

### 3.5 Indirect estimation of oceanic heat transports

We indirectly estimate the vertically integrated oceanic heat transport at a specific latitude of interest $\varphi$ in the North Atlantic
basin using the oceanic heat budget equation in the following form

$$OHT_\varphi = OHT_{\varphi_C} - \left[ F_S - \rho_0 c_p \frac{\partial}{\partial t} \int_0^Z (T_o - T_{ref}) dz \right]_\varphi^{\varphi_C} - R \Big|_{global} \tag{5}$$

where $OHT_{\varphi_C}$ is the heat transport through the choke point $\varphi_C$ (from DS+FS+BSO mooring-derived estimates; see red
lines in Fig. 1), and the second term on the right side describes the temporal ocean heat content tendency (OHCT; from 0–
2000m IAP data) subtracted from the net surface heat flux (from ERA5 forecasts) and averaged over the ocean area between
$\varphi_C$ and $\varphi$ (the Mediterranean Sea is excluded). This is the same approach as used by Trenberth and Fasullo (2017), Mayer
et al. (2022), and Baker et al. (2022) to indirectly estimate heat transports through the RAPID [Johns et al. (2011); McCarthy
et al. (2015); Bryden et al. (2020); see also Fig. 1 in Mayer et al. (2022)] and SAMBA array, respectively. $OHT_\varphi$ is calculated
for every fifth latitude between 0–60° N ($\varphi_C$ is situated between 67–80° N). Additionally, we adjust heat fluxes in the North
Atlantic basin by subtracting the monthly difference between global ocean mean vertically integrated 0–2000m OHCT and





$F_S$ (denoted as $R$) from each grid point. This removes inconsistencies between surface heat fluxes and OHCT and guarantees
temporal consistency in a global manner [see also Trenberth et al. (2019); Liu et al. (2020); Mayer et al. (2022)].

To indirectly estimate the AMOC trend over the whole study period, we extend the 2005–10 OHT climatology of the
choke point DS+FS+BSO to 1950-2019, under the assumption that the upward OHT trend at high latitudes is relatively weak
(Muilwijk et al., 2018; Wang et al., 2019, 2020; Docquier and Koenigk, 2021) compared to trends at low latitudes and basin-
wide heat flux trends. Trends of indirectly estimated OHT are computed in two ways, with monthly data and 5-year means,
using the procedure described by Loeb et al. (2022) (see section 3.2 therein) to estimate trend uncertainties; that is, the effective
sample size takes into account all significant autocorrelation functions $\rho$ up to lag m where $\rho_{m+1} < 0$ and $\rho_{m+1} + \rho_{m+2} < 0$
is satisfied. If this conditions are not satisfied for the autocorrelation at any lag, the true instead of the effective sample size is
used to estimate uncertainties.

### 3.6 Computation of climate indices

The AMO index is calculated similarly to the approach suggested by Trenberth and Shea (2006), where global-mean SST
anomalies are subtracted from the spatially averaged SST time series of the North Atlantic basin (0–60° N and 0–80° W). The
NAO index is derived from an EOF analysis applied to monthly surface pressure fields from ERA5 between 20–80° N and
90° W–40° E. The normalized principle component of the first EOF then describes the NAO index (Hurrell, 1995; Hurrell and
Deser, 2009).

## 4 Results

In the following, we split the study period 1950–2019 equally and consider the periods 1950–84 and 1985–2019 separately as
inferred air-sea heat fluxes from Mayer et al. (2022) are available only for the period from 1985 onward. Moreover, it has been
shown that the global warming trend has accelerated in the past few decades (Cheng et al., 2017; Fox-Kemper et al., 2021)
making the separation into two periods reasonable.

### 4.1 1985–2019 trends

Air-sea heat fluxes in the North Atlantic ocean exhibit a distinct annual cycle, with the largest ocean heat loss to the atmosphere
in boreal winter and strongest heat gain in summer. In boreal winter, fluxes are widely negative (heat loss from the ocean to
atmosphere) over the ocean basin, in particular at high latitudes and along the Gulf Stream where immense amount of oceanic
energy is transported northward and long-term averages of net surface heat fluxes can be as large as -400 W m$^{-2}$ (Fig. 1).
During summer, heat fluxes are positive (the ocean gains energy from the atmosphere) across the ocean basin, with values
ranging from zero (along the Gulf Stream and tropical North Atlantic) to 200 W m$^{-2}$ in coastal areas of North America and
Africa (not shown). Here, we focus on winter-month (December–February) heat fluxes as they feature the most pronounced
heat flux trends among all calendar months.





Over the past 35 years (1985–2019), several prominent regions with significant positive or negative trends have emerged
(Fig. 2a). Negative surface flux trends (stronger loss of energy from the ocean to atmosphere) can be found along the Gulf
Stream, and in regions of strong sea ice retreat that is driven by recent global warming (Fox-Kemper et al., 2021), e.g., along
the East Greenland current, in the Buffin Bay and Labrador Sea, and in the northern part of the Barents Sea. The retreat allows
the ocean to cool in areas that were otherwise covered by sea ice resulting in strong negative heat flux trends. Surface heat loss
in the tropical North Atlantic also strengthens significantly, but to a lesser extent than in the Gulf Stream or region of strong
sea ice retreat.

Positive trends (weakening of negative net surface heat fluxes during winter months) are prominent in the Norwegian and
Labrador Sea, but also in the region where Gulf Stream water masses bifurcate and form the North Atlantic Drift Current
further north and the equatorward propagating Azores current in the south (between 40°–50° N and 45°–25° W). This region
of strongly positive trends appears spatially more extended for inferred surface heat fluxes from Mayer et al. (2022) (see
Fig. 2b), but with similar peak value of about 29 W m$^{-2}$ dec$^{-1}$ (dec = 10 years). At other locations of the North Atlantic
Ocean, both flux products exhibit qualitatively similar trends indicating that ERA5 flux trends seem reliable in terms of spatial
structure, at least for the chosen study area and period of time. However, note that the trends in many areas are statistically
insignificant (e.g., the positive trends in the Labrador and Irminger Sea, or at the southern flank of the Gulf Stream) and thus
should be treated with caution when interpreting them.

Main contributor to the $F_S$ trend are turbulent heat fluxes (THF; Fig. 2c), whereas trends in radiative fluxes (RHF; Fig. 2d)
are usually an order of magnitude weaker (except for the Arctic ocean, which is not further discussed here). Spatial means
over the whole study area are -1.6 W m$^{-2}$ dec$^{-1}$ for THF and ∼0.1 W m$^{-2}$ dec$^{-1}$ for RHF resulting in negative $F_S$ trends of
about -1.4 W m$^{-2}$ dec$^{-1}$ during 1985–2019 (see Table 1). For comparison, inferred $F_S$ exhibits a weak positive trend of 0.3
W m$^{-2}$ dec$^{-1}$ owing to the spatially more extended positive trends.

We also computed mean trends of globally adjusted $F_S$ (see section 3) and net surface heat fluxes from Liu et al. (2020)
(also known as DEEP-C dataset; publicly available at https://doi.org/10.17864/1947.000347 for the period 1985–2017) over
the whole study area. The DEEP-C product is based on our inferred $F_S$ but unrealistic surface fluxes over land are subsequently
redistributed to the ocean, which removes spurious trends in the late 1990s and 2000s (Liu et al., 2017, 2020; Mayer et al.,
2022). As a consequence, $F_S$ from DEEP-C exhibits a realistic global ocean mean that matches the observed mean ocean heat
uptake and is thus well suited as reference for long-term trend studies (Mayer et al., 2022). For 1985–2017, we find a mean
trend of 1.4 W m$^{-2}$ dec$^{-1}$ for globally adjusted $F_S$ and 1.1 W m$^{-2}$ dec$^{-1}$ for the DEEP-C product (both are statistically
insignificant), indicating good agreement of the two estimates. For comparison, the unadjusted model-based $F_S$ exhibits a
1985–2017 mean trend of -1.6 W m$^{-2}$ dec$^{-1}$ indicating that the global adjustment of $F_S$ yields more realistic and reliable
trend estimates. This adds confidence to our globally adjusted $F_S$ data and its use for the full period starting in 1950.

In addition, we show each component of THF separately. In general, THF trends (and consequently also model-based $F_S$
trends) are governed by changes in latent heat flux (Fig. 2e) at low latitudes and sensible heat flux trends (Fig. 2f) at mid- and
high-latitudes (north of ∼40° N). Of the -1.6 W m$^{-2}$ dec$^{-1}$ mean THF trend, about -1.9 W m$^{-2}$ dec$^{-1}$ stem from latent heat





and 0.3 W m$^{-2}$ dec$^{-1}$ from sensible heat flux trends (Table 1). Along the sea ice edge, both components contribute equally to the negative THF trend as both were substantially lower when ocean areas were covered by sea ice before.

While flux trends shown in Fig. 2 are from forecasts, the following evaluations are based on analyzed quantities as they are better constrained by observations than their forecast counterpart. Most differences between forecast and analyzed flux trends can be related to moisture analysis increments (discussed below).

    Trends in latent and sensible heat fluxes can further be formally attributed to changes in 10 metre horizontal wind speed and temperature or humidity differences between lowest model level and ocean surface [see bulk formulae Eq. (1) and (2)].

Model-level humidity (Fig. 3a) uniformly increases at almost all locations as expected from a warming atmosphere [a warmer atmosphere can hold more moisture; Douville et al. (2021)]. The statistically insignificant decline in the eastern North Atlantic and Mediterranean Sea can be attributed to stronger northerly winds (see Fig. 4a) and declining moisture transport into that area as related to a strengthened NAO.

    Changes in model-level temperature (Fig. 3b) are qualitatively similar to those in atmospheric moisture, but are statistically

significant in almost all parts of the tropical North Atlantic. As the ocean warms due to climate change (Fox-Kemper et al., 2021), surface saturation humidity and skin temperature (Fig. 3c and d) increase almost everywhere (note that the surface saturation humidity is derived from skin temperature and surface pressure according to the Clausius-Clapeyron equation). The moderate and statistically insignificant decreasing trend in the Irminger Sea, which also appears weaker in model-level parameters, is a result of the anomalously cool ocean in the North Atlantic Warming Hole (Rahmstorf et al., 2015).

The surface fluxes are not so much governed by individual parameters at surface and model level, but by their differences from which several observations can be made:

    1. The trend pattern of ($q_{ml} - q_{sfc}$) and ($T_{ml} - T_{skin}$) are almost identical to that of latent and sensible heat fluxes (cf. Fig. 2e and f; pattern correlations are $> 0.8$) indicating that the horizontal wind speed (Fig. 4a) has a comparatively small impact on the spatial distribution of LHF and SHF trends.

2. Long-term changes in surface saturation humidity (governed by skin temperature trends) are in most areas of the North Atlantic stronger than changes in model-level humidity, and vice versa for the temperature. This results in almost uniformly negative ($q_{ml} - q_{sfc}$) but positive ($T_{ml} - T_{skin}$) trends. While surface saturation humidity increases with increasing skin temperature according to the Clausius-Clapeyron relation (i.e., relative humidity remains 100 %), the increase in model-level humidity is much weaker so that relative humidity decreases (Fig. 4b), especially south of 40° N. This means,

near-surface air masses in the tropical North Atlantic become drier relative to the temperature increase (the Clausius-Clapeyron relation would postulate stronger humidity trends for constant relative humidity), which can be caused by several factors (discussed below).

    3. Among the regions of strong sea ice retreat, peak positive trends in surface humidity and temperature can be found along the Gulf Stream region (with values up to 0.5 g kg$^{-1}$ dec$^{-1}$ and 0.8 K dec$^{-1}$). This leads to remarkably strong negative

trends in ($q_{ml} - q_{sfc}$) and ($T_{ml} - T_{skin}$) highlighting the importance of the ocean in this area. It should be noted that the




Gulf Stream signal is barely visible in relative humidity trends (Fig. 4b) due to a well-mixed boundary layer and strong coupling between atmosphere and the underlying warm Gulf Stream.

4. Over the North Atlantic Warming Hole, changes in the model-level temperature and humidity closely follow the Clausius-Clapeyron relation so that the warming hole signal is barely visible in RH trends (Fig. 4b). Trends in model-level and surface parameters are of similar strength resulting in weak and statistically insignificant heat flux trends (note that the North Atlantic Warming Hole is further north to the bifurcation area of the Gulf Stream and does not coincide with peak positive trends).

5. The strong positive ($T_{ml} - T_{skin}$) trend in the Norwegian Sea originates from positive trends in the atmosphere and somewhat less positive trends (or even negative trends east of Iceland) of the skin temperature. This can be attributed to trends towards more south-easterly winds (see Fig. 4a) advecting warmer air masses from lower latitudes to the Norwegian Sea, which is related to a strengthened Icelandic low.

6. In the Labrador and Nordic Seas, ($T_{ml} - T_{skin}$) trends downwind to areas of strong sea ice retreat become widely positive (mean wintertime climatology is a northerly wind in both basins; not shown). One explanation could be that air masses that are advected from further north get heated by the enhanced fluxes where sea ice retreated. The anomalously warm air-masses damp air-sea fluxes further south resulting in largely compensating sensible heat fluxes along the wind direction (spatial average is $\sim$2 W m$^{-2}$ dec$^{-1}$ over the Nordic Seas; see Table 1), with negative trends in areas of strong sea ice retreat and positive trends downwind.

One possibility to throttle the growth in near-surface humidity in the tropical North Atlantic is a stronger advection of dry air masses through intensification of the Hadley Cell. To manifest this, we present DJF trends of the zonally averaged meridional mass stream function derived from ERA5 wind fields (Fig. 5). The dipole structure between 0–30° N indicates that the north hemispheric Hadley cell has shifted poleward and strengthened in intensity, which enhances the subsidence of dry air along the northern flank of the Hadley Cell. This also agrees with positive trends in 10 metre wind speed between $\sim$20–30° N (Fig. 4a). Note that the mass stream function is obtained by integrating over all longitudes, and the intensification may take place over other ocean basins. However, a statistically significant increase of low-level cloud cover and outgoing longwave radiation (not shown) in the almost entire tropical North Atlantic Ocean indicate that the Hadley cell intensification also appears over the tropical Atlantic Ocean.

On the other hand, Trenberth et al. (2011) and Mayer et al. (2021) noted that analysis increments introduced by the data assimilation system due to changes in the observing system can artificially remove or add atmospheric moisture, which in turn could influence near-surface humidity trends. To investigate the impact of atmospheric moisture and temperature increments in the lowest model level on air-sea heat fluxes from ERA5, we compute latent and sensible heat fluxes according to the bulk formulas (Eqs.1 and 2) using both forecast and analyzed state quantities (not shown). Differences between trends derived from analyses and forecasts can then be used as rough estimate for trend uncertainties caused by analysis increments (note that only temporally varying analysis increments introduce an artificial trend, and not a constant offset between analyses and forecasts).



We find strongest variation in humidity increments (approximated by $q_{an}$-$q_{fc}$) in the tropics, with 1985–2019 trends up to
$\sim$0.05 $\mathrm{g\,kg^{-1}\,dec^{-1}}$ and values of -0.2 $\mathrm{g\,kg^{-1}}$ in the late 2000s and early 2010s (see appendix B). Humidity increments in
earlier times and at higher latitudes are in general less negative. Temperature increments are temporally more stable and almost
independent of latitude, with values ranging between $\pm$0.03 $\mathrm{K\,dec^{-1}}$, and are thus less impactful on turbulent heat fluxes.
The latent heat flux trends derived from analyses are in the zonal mean of the North Atlantic 1–2 $\mathrm{W\,m^{-2}\,dec^{-1}}$ stronger than
those based on forecasts (root mean square error between trends derived from analysis and forecasts is 1.1 $\mathrm{W\,m^{-2}\,dec^{-1}}$ over
the ice-free ocean). The negative humidity increments in the lowest model level artificially remove moisture from the model,
which results in larger ($q_{ml}$-$q_{sfc}$) differences and thus stronger analyzed latent heat flux trends. For sensible heat fluxes, analysis
increments are less important. Zonally averaged SHF trends derived from analyses are <0.4 $\mathrm{W\,m^{-2}\,dec^{-1}}$ larger than those
derived from forecasts (RMSE over the whole study area is 0.5 $\mathrm{W\,m^{-2}\,dec^{-1}}$). Therefore, we argue that the regional impact of
all relevant analysis increments introduced by the ERA5 data assimilation on air-sea heat flux trends is rather small during the
1985–2019 period. While trends from analyses are at all latitudes stronger than those derived from forecasts, the spatial pattern
of the trends remains almost unaffected (i.e., the difference between analyses and forecasts is smaller for weaker trends, and
vice versa); that is, the negative LHF trend in the tropical North Atlantic is most likely a result of Hadley cell intensification
and can not be explained by temporally varying moisture increments.

In summary, long-term changes in net surface heat fluxes over the North Atlantic Ocean are primarily driven by latent heat
flux trends (Table 1), which are associated with changes in the surface (related to changes in skin temperature) and model-level
humidity (e.g., advection of drier air masses), while changes in wind speed are negligibly small. Furthermore, we conclude
that temporally varying analysis increments influence the magnitude of air-sea heat flux trends by about 1–2 $\mathrm{W\,m^{-2}\,dec^{-1}}$,
whereas their spatial pattern remains widely unchanged.

### 4.2   1950–1984 trends

Long-term latent and sensible heat flux changes before 1985 (Fig. 6) differ in most areas substantially from those during the
more recent period. Most notable are the strong positive and significant LHF trends in the Caribbean Sea, along the Gulf Stream
north of $\sim$40° N, and in the Labrador Sea.

Trends in sensible heat flux are strongest along the sea ice edge and are absent in the Norwegian Sea, the region with the
strongest weakening after the 1980s (cf. Fig. 2f). The widely positive trends can be attributed to a stronger temperature decrease
at the surface relative to the atmosphere (not shown). We find remarkably strong correlations (0.5–0.8) between trends of skin
temperature and 0–300m OHC from IAP for all four box averages (see Fig. 1) suggesting that the basin-wide weakening of
sensible heat fluxes is related to the ocean cooling that occurred during that time [see Hodson et al. (2014)]. Consequently, there
is no warming hole signal in the skin and model-level temperature, which is consistent with results from Chemke et al. (2020)
based on satellite-based HadISST data [see supplementary information therein; in fact, ERA5 employs the second version of
this dataset as SST forcing, see Hersbach et al. (2020)]. Temperature analysis increments before 1985 do not play an important
role due to their negligibly weak trends of less than $\pm$0.02 $\mathrm{K\,dec^{-1}}$ in ice-free regions. This results in differences between





trends derived from analysis and forecast data of less than $0.5\ \mathrm{W\,m^{-2}\,dec^{-1}}$ in the zonal mean, with an RMSE of about 0.5 $\mathrm{W\,m^{-2}\,dec^{-1}}$ for the ice-free ocean.

In accordance with the ocean cooling, surface saturation humidity decreases almost everywhere (not shown), with the strongest negative trends along the Gulf Stream. The only larger patch of positive (but statistically insignificant) trends appear in the subtropics and along the sea ice edge. Near-surface humidity also decreases where temperature decreases, but weaker than the CC-related decrease such that the relative humidity increases significantly in most areas (whereas negative trends in all ice-free areas are insignificant; not shown). This leads to mostly increasing humidity differences ($q_{ml}$-$q_{sfc}$) and thus also in positive latent heat fluxes trends (Fig. 6a).

As for the SHF, we exclude analysis increments as possible source of uncertainties during that time because of their weak trend of $\pm 0.02\ \mathrm{g\,kg^{-1}\,dec^{-1}}$ over most locations of the ice-free ocean. Therefore, differences between LHF trends derived from analyses and forecasts are less then $0.6\ \mathrm{W\,m^{-2}\,dec^{-1}}$ in zonal mean before 1985 (RMSE over the ice-free ocean is 0.5 $\mathrm{W\,m^{-2}\,dec^{-1}}$). Changes in 10-metre wind speed are mostly insignificant and of similar strength as after 1985, and thus affect turbulent heat fluxes only marginally.

### 4.3 Flux trends in focus regions

To understand long-term changes in the four thermodynamically interesting areas of the North Atlantic (see boxes in Fig. 1) in more detail, we show spatial averages of model-based and inferred $F_S$, and partial trends of analyzed input variables as regressed onto LHF and SHF from ERA5 forecasts, for the period 1950–2019 (Fig. 7). In the Norwegian Sea (NWS), air-sea heat fluxes weaken particularly in the late 2000s and 2010s (Table 1), which is in good agreement with the enhanced oceanic heating during that time (Mork et al., 2019; von Schuckmann et al., 2021). However, the advection of warmer more humid air associated with changes in 10 metre wind direction appears to overcompensate oceanic trends (right panel in Fig. 7; partial trends of surface quantities are negative as they contribute with opposite sign to turbulent heat flux trends) so that LHF and SHF trends are relatively weak compared to those in model-level or surface quantities alone. Note that Skagseth et al. (2020) found similar changes in wind direction for the adjacent Barents Sea.

Long-term changes in the North Atlantic Warming Hole (NAWH) are the weakest among the four areas of interest. Derived trends should thus be treated carefully as they also depend strongly on the chosen reference period. To test the robustness of trends in the four study areas, we considered various reference periods. For instance, we removed the last year from the time series and computed DJF trends based on 1950–2018. While fluxes steadily increase in the NWS and decrease in the tropical North Atlantic (TNA) almost independently of the chosen reference period, trends in the NAWH and GS region are less than $1\ \mathrm{W\,m^{-2}\,dec^{-1}}$ (or less then $2\ \mathrm{W\,m^{-2}\,dec^{-1}}$ when considering 1950–2019, see Table 1) and statistically insignificant. From this, we cautiously argue that heat fluxes in the NAWH, and also in the GS box where trends from the early and late period compensate each other, do not exhibit a prominent long-term trend over the past 70 years as related to global warming, while changes in the TNA and NWS are most likely a result of global warming.

We also explore the winter-month $F_S$ climatology along the Gulf Stream extension on decadal timescales in order to reveal any signal in air-sea heat fluxes associated with a poleward displacement due to global warming. Besides an oscillatory





behaviour similar to temporal changes in the more regional GS box shown in Fig. 7, we could not find a distinct sign of a poleward shift in air-sea heat fluxes, which is consistent with findings from Yang et al. (2016).

The partial trends in Fig. 7 show that trends in model-level and surface quantities almost always act in opposite direction and thus compensate each other to some degree (except for moisture in the TNA during the first period, where both are negative).
In addition, it demonstrates qualitatively that the impact of 10 metre wind speed on heat flux trends is rather small compared to changes in moisture or temperature, especially in cases where latent or sensible heat fluxes exhibit trends of several Watts per square metre.

We also find that in the three northernmost boxes, trends of inferred and model-based fluxes have the same sign but differ by about 2–4 $\mathrm{W\,m^{-2}\,dec^{-1}}$. Inferred fluxes exhibit stronger upward trends in the NWS and NAWH, and a weaker downward
trend in the GS area (Table 1). Trends in the TNA coincide remarkably well underlining the reliability of model-based fluxes from ERA5 forecasts in that particular region. In summary, this suggests that model-based trends are largely reliable in terms of sign and spatial pattern (see also Fig. 2).

### 4.4 Long-term impact of natural variability modes

The North Atlantic Oscillation (NAO) is a periodic oscillation in sea level pressure and wind (Visbeck et al., 2001) and can
temporally and regionally influence air-sea interactions. During the last 30–40 years, the NAO tends to more positive phases (strengthened Icelandic low and Azores high) than before, which has been attributed to global warming (Gillett et al., 2003). Here, we want to explore its long-term impact on trends of air-sea heat fluxes (Fig. 8).

Long-term $F_S$ changes over the entire study period appear to be weaker and spatially more uniform compared to those
over the two sub-periods discussed before. Trends are widely positive in the western North Atlantic, in the region of the North Atlantic Warming Hole, and in the Norwegian Sea. Persistent negative flux trends occur in the tropical North Atlantic, along the Gulf Stream, and in regions of strong sea ice retreat which are largely consistent with negative changes during both sub-periods.

The December–February NAO regressed onto $F_S$ features a basin-wide tripolar pattern (see appendix C), with strong negative values in the Irminger and Labrador Sea, negligibly weak trends in the tropical and subtropical latitudes, and positive
values in between (with peak values along the Gulf Stream). The more frequent occurrence of positive NAO phases over the past 30 years, relative to 1950–90, seem to favour anomalous ocean cooling at higher latitudes and heating in the western North Atlantic (Fig. 8a). Removing the NAO signal from $F_S$ trends (Fig. 8b) thus weakens ocean cooling (more positive trends) in the Irminger and Labrador Sea over time and allows stronger cooling in the western North Atlantic (less positive trends). In addition, we find weak correlations of less than 0.4 between NAO index and $F_S$ box averages in the TNA, GS, and NWS, but
-0.75 for the NAWH. This indicates that the $F_S$ trend over the North Atlantic Warming Hole box is strongly influenced by trends of the NAO and its tendency toward more positive phases, while other areas are less effected.

Despite the remarkably strong regional impact of the NAO on air-sea heat fluxes at high latitudes, its spatial mean averaged over the whole study area is less than 0.03 $\mathrm{W\,m^{-2}\,dec^{-1}}$ (1950–2019). For comparison, the 1950–2019 $F_S$ trend averaged over the whole study area (as shown in Fig. 8a) is 0.14 $\mathrm{W\,m^{-2}\,dec^{-1}}$. This suggests that the trend toward more positive NAO





phases only leads to a relocation of areas where oceanic heat is lost or taken up through surface fluxes, rather than a steady

increase in anomalous ocean heat uptake as related to global warming. This somewhat agrees with the finding of Cohen and

Barlow (2005), that the global DJF warming trend during 1972–2004 may be unrelated to regional warming trends driven by

the NAO.

We also regressed the AMO forcing (Kerr, 2000) onto $F_S$ to estimate its long-term impact on flux trends (see appendix

C). The AMO partial trend varies between $\pm 2$ W m$^{-2}$ dec$^{-1}$ over the ice-free ocean, with negative values in the Irminger

and Labrador Sea and around the North Atlantic Warming Hole (40–60° and 50–20° W), and positive values elsewhere.

Although the AMO impact on flux trends in the Irminger and Labrador Sea has the same sign and similar spatial structure as

the NAO forcing, its strength over the 70-year period is weaker. Additionally, we find a spatial mean of the AMO signal of 0.22

W m$^{-2}$ dec$^{-1}$, which points to a basin-wide weakening of air-sea heat fluxes, but this is likely an effect of non-zero AMO

trend due to the relative shortness of the time series (the AMO does not complete a full period during the whole study period),

and is likely not related to global warming.

## 4.5  Changes in the Atlantic Meridional Overturning Circulation

In the previous sections we have diagnosed a reduction of the net surface heat flux from ocean to atmosphere during 1950–

2019 when averaging over the North Atlantic. This reduction could be related to a cooling trend of the underlying ocean and/or

a reduction of oceanic heat transports associated with the AMOC. In this section we explore both possibilities to verify the

AMOC trends with observation-based data (reanalysis is a combination of observations and forecasts). The ocean heat transport

at different latitudes of the North Atlantic basin is indirectly estimated from the ocean heat budget using globally adjusted $F_S$

from ERA5 forecasts and OHC data from IAP (see section 3). Here we focus on full-year OHT estimates because it increases

the signal-to-noise ratio (sub-annual OHC changes are often related to seasonally compensating trends in wind patterns) and

observational uncertainties of OHCT are considered larger on sub-annual time scales. Furthermore, annual mean $F_S$ trends

are similar to seasonal DJF trends in terms of spatial pattern (pattern correlation is ~0.8), but are generally weaker across the

North Atlantic basin (root mean square of trends is 1.8 W m$^{-2}$ dec$^{-1}$ as compared to 3.7 W m$^{-2}$ dec$^{-1}$).

Results for 0–60° N are shown in Fig. 9 using two types of trend estimates (see section 3). Both estimates show more negative

trends (weakened AMOC) and larger uncertainties at lower latitudes, with a maximum at the equator. While the method based

on five-year means gives significant trends for all latitudes except 35–50° N (averaging over 5 years reduces the variance),

linear regression on monthly data is statistically significant only between 45–60° N.

Main contributor to the weakened OHT in the North Atlantic basin are statistically significant long-term changes of glob-

ally adjusted air-sea heat fluxes, whereas the trend of meridionally integrated OHCT is comparably small and insignificant

throughout all latitudes between the equator and 60° N (see Table 2 for trends integrated over the area between choke point

DS+FS+BSO and 26° N). Removing the OHCT term from the budget equation [integral term in Eq. (5)] thus reduces trend un-

certainties while the strength and meridional structure of the estimated OHT trend remain roughly the same (compare left and

middle panel of Fig. 9). In other words, the AMOC weakening is primarily associated with a positive trend of globally-adjusted

$F_S$ and thus a decline of ocean-to-atmosphere heat fluxes (1950–2019 mean is -13.7 W m$^{-2}$). We also computed the indirectly





estimated OHT trend based on the sub-periods 1950–84 and 1985–2019 but could not find a significant AMOC weakening in

either period (not shown).

## 5    Summary and discussion

In this work, we investigated the reliability and temporal stability of winter-months (December–February) trends of model-based net surface heat fluxes from ERA5 forecasts over the North Atlantic Ocean during 1950–2019. Main drivers of these trends are identified using analyzed state quantities from ERA5, and the influence of natural variability modes and analysis

increments as introduced by the ERA5 data assimilation system are considered. Whenever possible, ERA5 forecast fluxes are compared with indirect estimates from Mayer et al. (2022), which are proven to be temporally stable and exhibit a small mean bias over the global ocean. Furthermore we performed a linear perturbation analysis on turbulent heat fluxes in four distinct $8 \times 8°$ boxes, which allowed to quantitatively attribute flux trends to changes in wind speed, moisture, and temperature, assuming a linear regime. In a final step, we used basin-wide annual mean air-sea heat fluxes to indirectly estimate the AMOC

trends over the past 70 years, and discussed its reliability and sources of uncertainties.

We find that air-sea heat flux trends at low (high) latitudes are largely driven by long-term changes of differences between model level and surface humidity (temperature). We further traced surface trends back to local changes in the ocean heat content, whereas model level trends strongly depend on altered conditions of advected air masses through changes in wind direction, and not so much in wind speed. This process likely plays a major role in the tropical North Atlantic where increasingly

drier air masses are advected (likely linked with a strengthening of the Hadley cell), as well as in the Norwegian Sea, where increasingly warmer air is advected.

A more quantitative assessment of turbulent heat fluxes in four individual sub-regions reveals that the relative contribution of wind speed to turbulent heat flux trends is indeed negligible, and that surface and model level trends largely compensate each other. Furthermore, it is shown that trends in the later period (1985–2019) are substantially stronger compared to the early

period, which is consistent with accelerated warming in the past few decades (Cheng et al., 2017; Fox-Kemper et al., 2021). It should be noted that the strength of trends clearly depends on the chosen averaging area, especially in the Gulf Stream where north-south gradients of surface flux trends are steep.

The long-term changes in air-sea heat fluxes could have some further implications on weather and climate. For instance, the increased intensity of tropical cyclones during the past 40 years (Kossin et al., 2020) could possibly be linked to stronger latent

heat fluxes in the tropical North Atlantic (Fig. 8a; similar trends can be found for the Hurricane season September–November). Similarly, the negative heat flux trends over the Gulf Stream are most likely a response to an increased storm frequency, which in further consequence favours more cyclogensis (Shaman et al., 2010).

We also examined the impact of NAO and AMO on long-term $F_S$ trends. The more frequent positive NAO phases during the last 30–40 years significantly alter trends at high latitudes. It favours stronger ocean heat loss to the atmosphere via air-sea

heat fluxes in the Irminger and Labrador sea and anomalously weak loss in the western North Atlantic, albeit the basin-wide mean heat exchange between atmosphere and ocean remains unaffected. The AMO forcing, on the other hand, is weaker than





the NAO forcing but exhibits a non-zero mean in the North Atlantic basin, but robust statements about the impact of AMO are difficult given the relative shortness of the considered time series.

Finally, we linked the basin-wide air-sea heat flux trend to the AMOC weakening found in other studies by evaluating the oceanic heat budget using surface fluxes from ERA5 forecasts, OHCT data from IAP, and ocean heat transport data from Arctic Gateways in the north (i.e., mooring-derived estimates from the Davis Strait, Fram Strait, and Barents Sea Opening; see Fig. 1). Trend estimates based on monthly data exhibit large uncertainties and are insignificant south of 45° N, whereas computations based on 5-year means yield significant trends at almost all latitudes (taking into account all significant autocorrelation coefficients, see section 3). Removing the OHCT term from the calculations reduces uncertainties while trends remain approximately the same (the long-term OHCT trend is small compared to that in $F_S$ but introduces noise). Based on these results, we provide new and independent evidence for a weakening of the AMOC over the past 70 years [see also Rahmstorf et al. (2015); Caesar et al. (2018); Fox-Kemper et al. (2021); Boers (2021)], which is associated with positive heat flux trends (weakened negative fluxes) in the North Atlantic basin. We argue that the mean ocean heat transport through the choke point in the Nordic Seas is small (∼0.15 PW) so that even relatively large changes would not have a strong impact on indirectly estimated OHT trends further south [see Muilwijk et al. (2018) for long-term simulations of ocean heat transports through Arctic gateways].

Analysis increments of moisture and temperature at model level (i.e., the difference between analysis and forecast) likely influence the strength of trends but not so much the basin-wide spatial pattern. At most locations, moisture is removed from the model by the assimilation process resulting in stronger trends from analysed data by about 1–2 W m$^{-2}$ dec$^{-1}$ as compared to forecast data. Strongest moisture analysis increments can be found in the tropics in the late 2000s. At higher latitudes and before 2000, moisture increments are temporally stable and have negligible impact on surface heat flux trends. Temperature increments are relatively small and stable throughout the study period and thus play only a secondary role.

In the early period, observations are temporally and spatially sparse resulting in analyzed states that are closer to the model climate (to which forecasts are drifting) than to observations. Over time, more and more observational data are assimilated pulling the analysis away from the model climate. This increases analysis increments, which can have several implications on air-sea heat flux trend estimates. When trends are weak or compensate each other such that signal-to-noise ratio becomes low (e.g., when averaging over large areas), analysis increments can have a relatively large impact on the trend estimate. For example, the heat flux trend in the tropical North Atlantic box (see Fig. 1) is only -2.7 W m$^{-2}$ dec$^{-1}$ during 1985–2019 (Table 1). Analysis increments increasingly remove moisture from the atmosphere in that region (see appendix B) so that the trend based on analyzed state quantities is -4 W m$^{-2}$ dec$^{-1}$ (not shown). This is a 50 % stronger trend compared to the forecast-based estimate. Nonetheless, it is important to note that this is still a factor of ∼3 smaller than the trend uncertainty listed in Table 1. A similar effect can be found for global ocean and basin-wide averages as used to estimate the AMOC weakening. Both suffer from temporal inconsistencies in the late 1990s and early 2000s, which are likely caused by changes in the atmospheric observing system and hence analysis increments. Nevertheless, given that trends in analysis increments are spatially relatively uniform, we find that the applied global correction removes much of the effect of spurious air-sea flux trend on our inferred estimate of OHT.



From our results, we find that analysis increments are a useful tool for interpreting the trend estimates based on reanalysis data. Air-sea heat flux trends from ERA5 forecasts in the North Atlantic basin seem reliable in terms of sign and spatial structure, but we speculate that temporal inconsistencies in the late 1990s and 2000s [as shown by Mayer et al. (2022) for global ocean averages] and temporally varying analysis increments have a common cause which is the increasing number of
observations that indicate a drier atmosphere than in the model climate. Further research is needed to fully understand their impact on both forecast and analysis-based trends.

## Appendix A: Linearized turbulent heat fluxes

Turbulent heat fluxes are linearized by decomposing each variable on the right side of Eq. (1) and (2) into a mean state (with overbar) and deviation from the mean (with prime); that is, we substitute $\rho = \overline{\rho} + \rho'$, $|U_{ml}| = \overline{|U_{ml}|} + |U_{ml}|'$, $\Delta q = \overline{\Delta q} + \Delta q'$,
and $\Delta T = \overline{\Delta T} + \Delta T'$, where $\Delta q = q_{ml} - q_{sfc}$ and $\Delta T = T_{ml} - T_{skin}$. After some calculus, turbulent heat fluxes can be separated into a non-linear and linear part, where the former contains all products with more than one deviation term (e.g., the non-linear term $\overline{|U_{ml}|}\,\rho'\,\Delta q'$; not shown). The linear latent heat flux can be written as

$$F_{LH,linear} = C_Q\, L_v\, \left( \overline{|U_{ml}|}\,\overline{\rho}\,\overline{\Delta q} + \overline{|U_{ml}|}\,\overline{\rho}\,\Delta q' + \overline{|U_{ml}|}\,\rho'\,\overline{\Delta q} + |U_{ml}|'\,\overline{\rho}\,\overline{\Delta q} \right), \tag{A1}$$

and the linear sensible heat flux as


$$F_{SH,linear} = C_H\, c_p\, \left( \overline{|U_{ml}|}\,\overline{\rho}\,\overline{\Delta T} + \overline{|U_{ml}|}\,\overline{\rho}\,\Delta T' + \overline{|U_{ml}|}\,\rho'\,\overline{\Delta T} + |U_{ml}|'\,\overline{\rho}\,\overline{\Delta T} \right)$$
$$+ C_{SH}\, g\, z\, \left( \overline{|U_{ml}|}\,\overline{\rho} + \overline{|U_{ml}|}\,\rho' + |U_{ml}|'\,\overline{\rho} \right). \tag{A2}$$

## Appendix B: Moisture and temperature increments

Figure B1 shows 1985–2019 trends of moisture and temperature increments and corresponding TNA box averages for the
whole study period. Note that moisture increments in the tropical North Atlantic before 2000 are remarkably stable around zero but rapidly decrease afterward, with minima values of about -0.2 g kg$^{-1}$ in 2010–15 (negative analysis increments mean that moisture is removed from the model by the data assimilation). Temperature increments show a weak increase in the early 1990s but are temporally stable between 0–0.1 K otherwise.

## Appendix C: NAO and AMO regression onto air-sea heat fluxes

Figure C1 shows winter-months partial trends of NAO and AMO as regressed onto air-sea heat fluxes from ERA5 forecasts for the period 1950–2019.





*Author contributions.* All authors participated in the discussion and conceptual design of the paper. JM prepared the figures and wrote the manuscript under the supervision of LH and MM.

*Competing interests.* The contact author has declared that none of the authors has any competing interests.

*Acknowledgements.* JM and MM were financially supported by the Austrian Science Funds (FWF) project P33177. LH received support from the Austrian HRSM project GEOCLIM.



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

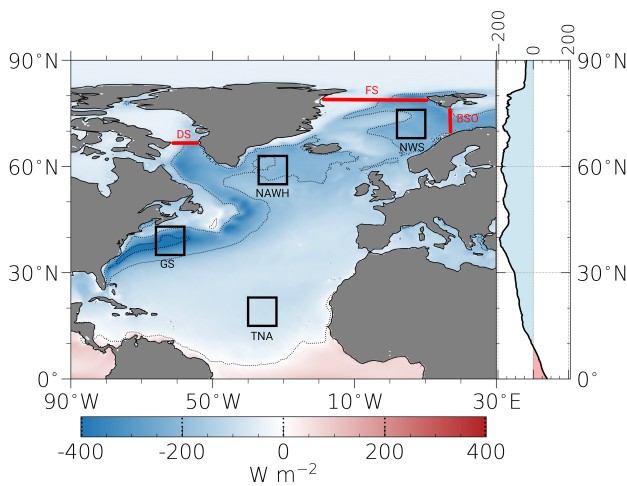

**Figure 1.** Mean 1985–2019 December–February climatology of model-based net air-sea heat fluxes from ERA5 forecasts. Black boxes indicate the four areas of interest located in the Norwegian Sea (NWS), North Atlantic Warming Hole (NAWH), Gulf Stream (GS), and tropical North Atlantic (TNA). The red lines mark the mooring locations in the Davis Strait (DS), Fram Strait (FS), and Barents Sea Opening (BSO), which are used to indirectly estimate the ocean heat transport in the North Atlantic basin. Contour lines are shown for 0, $\pm 200$, and $\pm 400$ W m$^{-2}$.

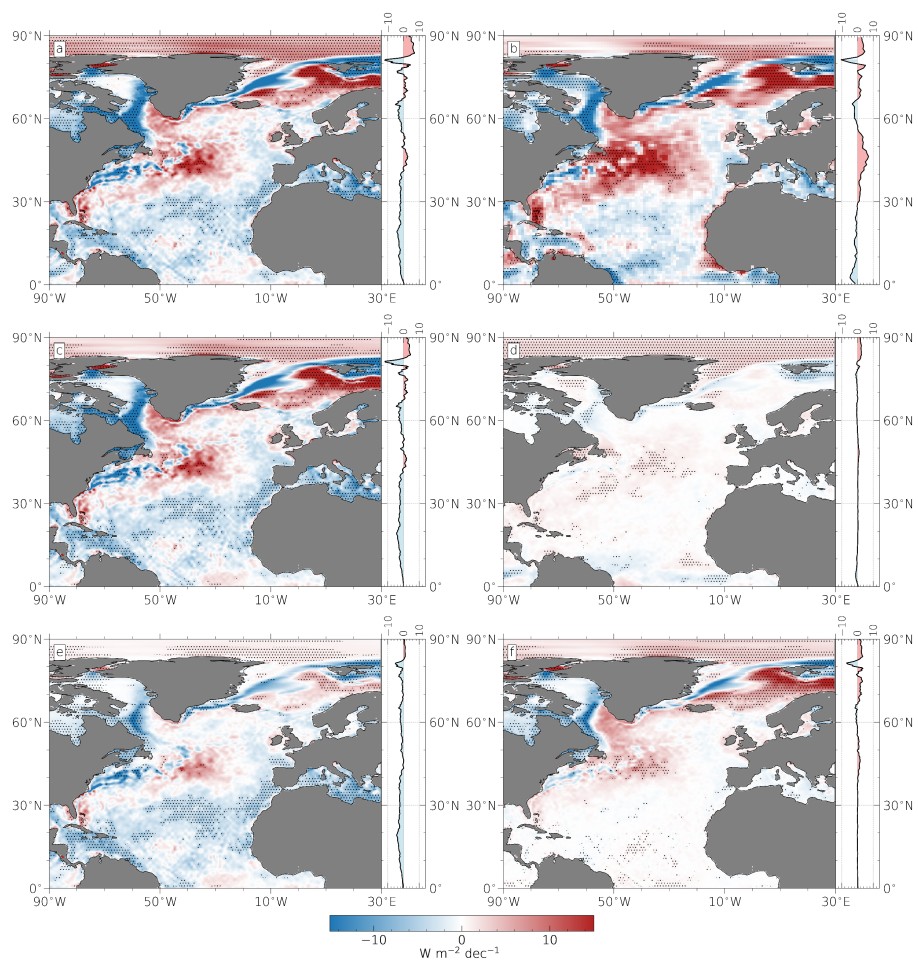

**Figure 2.** Linear trends of a) model-based net surface heat fluxes from ERA5 forecasts and b) inferred heat fluxes derived from atmospheric energy transports and TOA radiation for the period 1985–2019. Panel c) and d) show the turbulent and radiative energy flux component of model-based $F_S$ trends, and panel e) and f) illustrate latent and sensible heat fluxes separately. All trends are computed based DJF means of anomalies. Units are W m$^{-2}$ dec$^{-1}$. The shading represents areas of statistically significant trends (95 % confidence level).

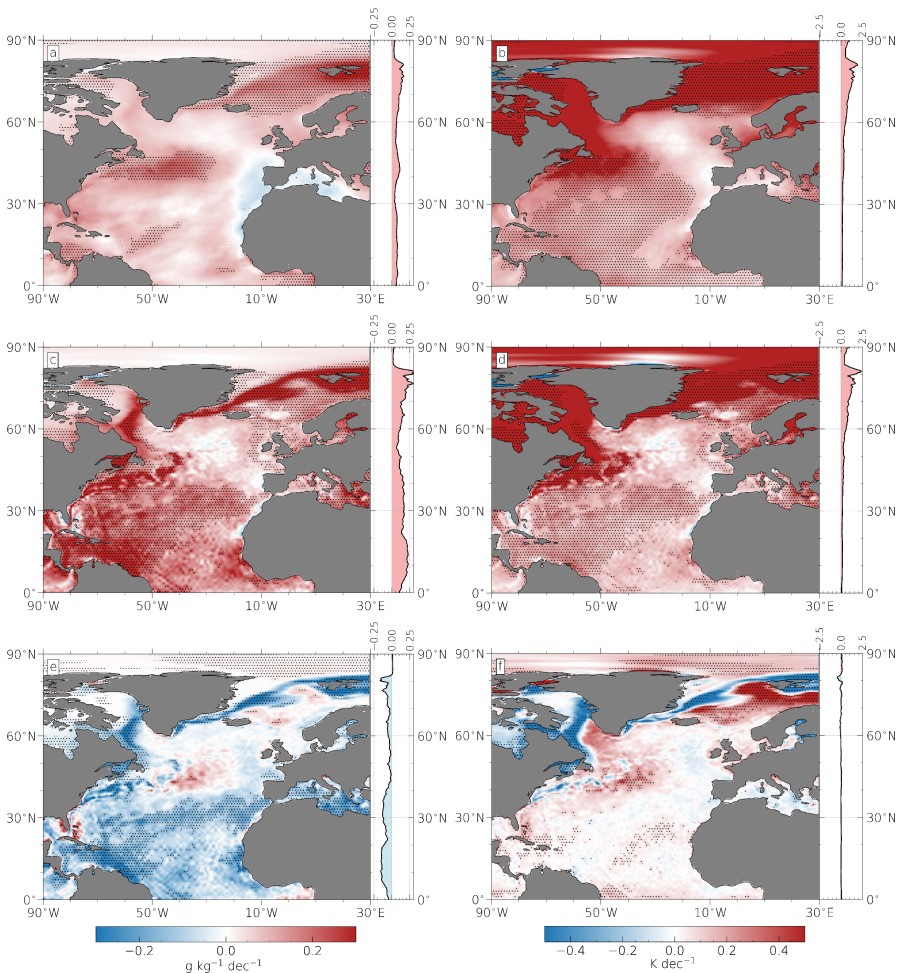

**Figure 3.** Linear DJF trends of analyzed a) model-level humidity, b) model-level temperature, c) surface saturation humidity, and d) skin temperature anomalies for 1985–2019. In addition, the difference between model-level and surface e) humidity and f) temperature is shown. Units are $\mathrm{g\,kg^{-1}\,dec^{-1}}$ for humidity trends and $\mathrm{K\,dec^{-1}}$ for temperature trends. The shading represents areas of statistically significant trends (95 % confidence level).

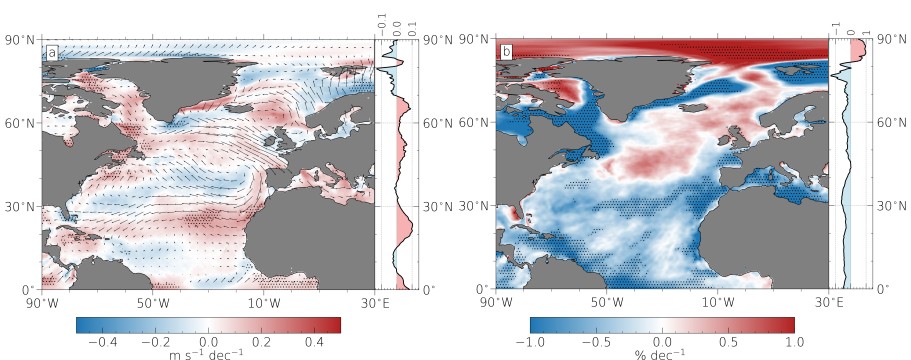

**Figure 4.** Linear trend of analyzed a) 10 metre horizontal wind speed and direction anomalies ($\text{m s}^{-1}\,\text{dec}^{-1}$) and b) model-level relative humidity ($\%\,\text{dec}^{-1}$) for DJF 1985–2019. Anomalous wind direction trends are illustrated by black arrows (with a maximum of $\sim$0.8 $\text{m s}^{-1}\,\text{dec}^{-1}$). The shading represents areas of statistically significant trends (95 % confidence level).



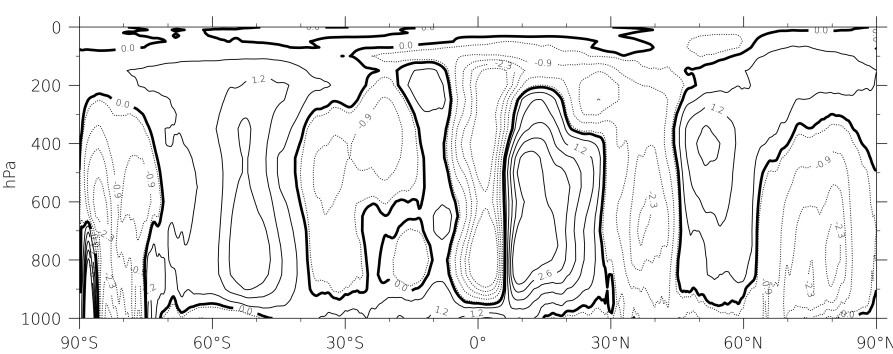

**Figure 5.** Linear trend of the meridional mass stream function from analyzed ERA5 winds over the period 1985 to 2019. Positive trends are shown as solid contour lines, negative trends as dotted lines. The stream function is integrated over 360 degrees in longitude. Units are $10^9$ $\mathrm{kg\,s^{-1}dec^{-1}}$.





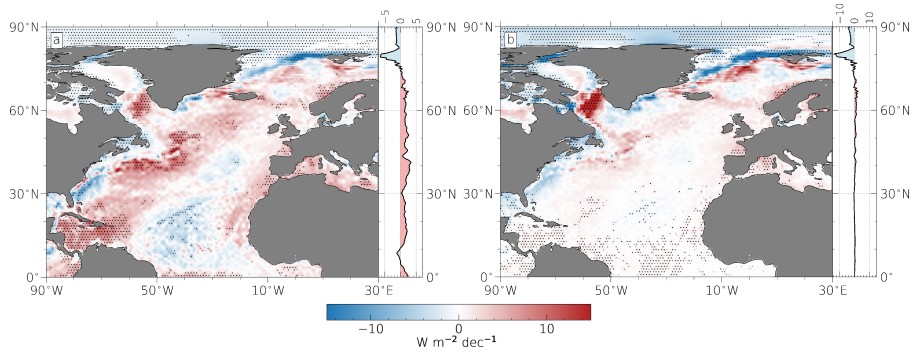

**Figure 6.** As in Fig. 2e and f, but for 1950–84.





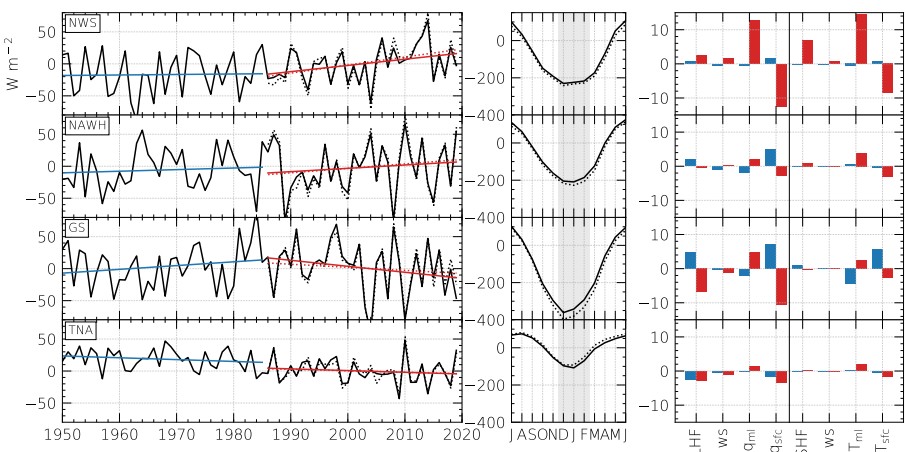

**Figure 7.** (left) DJF anomalies and (middle) full-year climatology of model-based $F_S$ from ERA5 forecasts (solid lines) and inferred $F_S$ (dotted lines) for box averages (see Fig. 1) in the Norwegian Sea (NWS), North Atlantic Warming Hole (NAWH), Gulf Stream (GS), and Tropical North Atlantic (TNA). (right) Partial trends [see Eq. (3)] of 10 metre wind speed (ws), model-level humidity ($q_{ml}$) and temperature ($T_{ml}$), surface saturation humidity ($q_{sfc}$), and skin temperature ($T_{sfc}$) as regressed onto latent (LHF) and sensible heat fluxes (SHF), respectively. Anomalies are computed w.r.t. 1985–2019. Trends for 1950–84 (1985–2019) are shown in blue (red). The grey background in the middle panel highlights the boreal winter months December–February. Units are $\mathrm{W\,m^{-2}}$ for anomalies and annual cycles, and $\mathrm{W\,m^{-2}\,dec^{-1}}$ for trends.





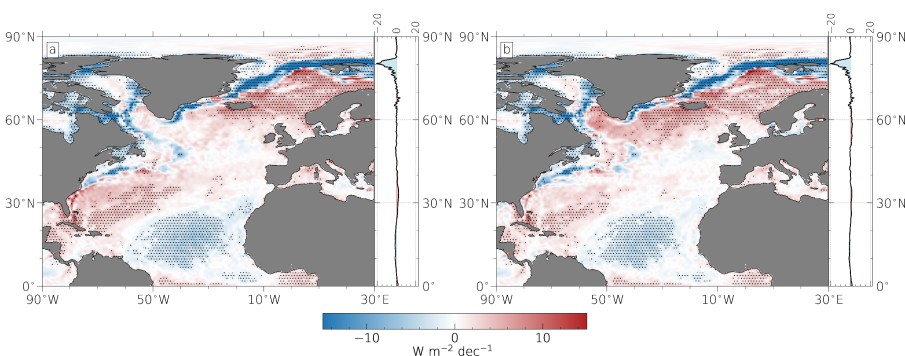

**Figure 8.** Linear trend of a) model-based surface heat fluxes from ERA5 forecasts for 1950–2019, and b) the same but with the partial NAO trend subtracted (see appendix C for regression pattern). The shading represents areas of statistically significant trends (95 % confidence level).





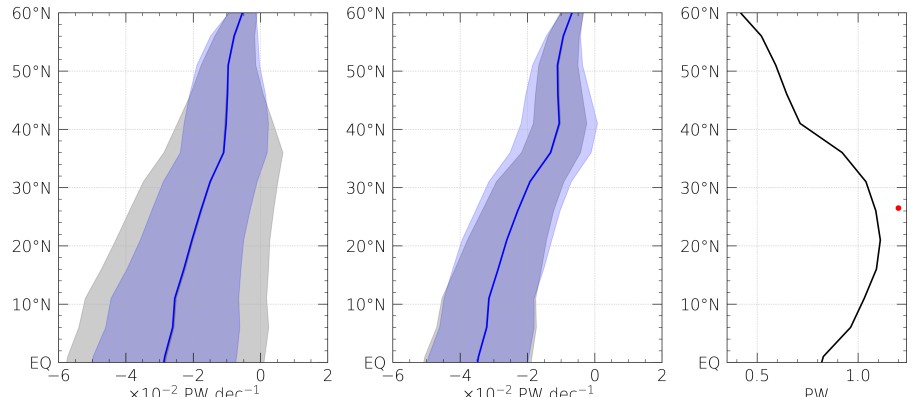

**Figure 9.** (left) 1950–2019 full-year trend of indirectly estimated Atlantic ocean heat transport as derived from the oceanic heat budget and (middle) the same but without OHCT data employed. Blue (grey) lines are trend estimates based on 5-year (monthly) means. The shading illustrates the 95 % confidence interval of the trend estimate. (right) 1950–2019 mean total indirectly estimated heat transport at each latitude. The red dot shows the 2004–2018 mean observed ocean heat transport through the RAPID array at 26.5° N.

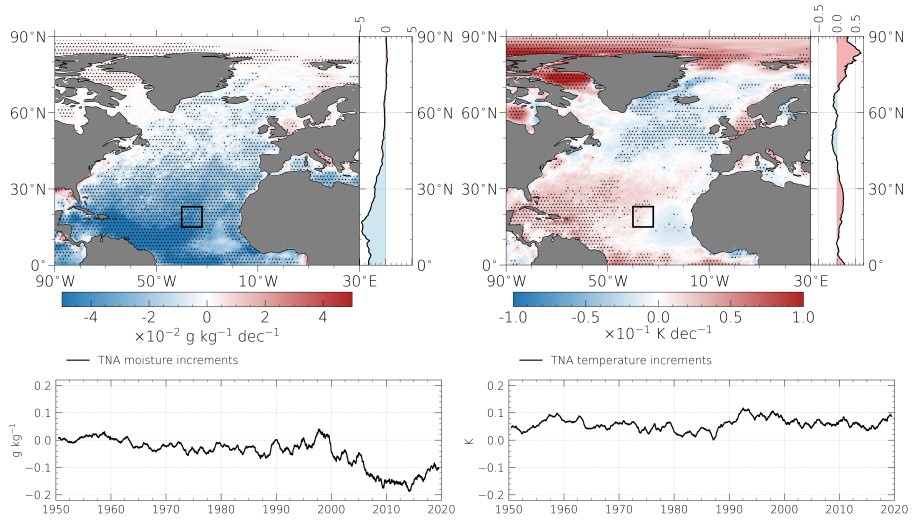

**Figure B1.** Analysis increments (analysis minus forecast field) of (left) moisture and (right) temperature at lowest model level. The upper panel shows DJF trend maps of analysis increments for the period 1985–2019. The lower panel shows analysis increments of the TNA box average for the whole study period. Time series are smoothed by a 12-month moving average.



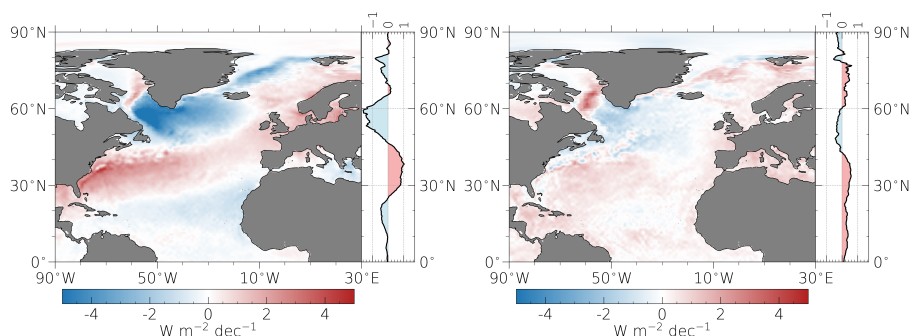

**Figure C1.** Partial trends of (left) NAO and (right) AMO as regressed onto air-sea heat fluxes from ERA5 forecasts for the period 1950–2019. Partial trends are computed for each grid point by multiplying the sensitivity between climate index and $F_S$ with the linear trend of the climate index [see explanation to Eq. (3)].





**Table 1.** Flux trends for various areas and periods of time. The study area refers to the ocean area between 0–90° N and 90° W–30° E. Nordic seas include the ocean area between 60–82° N and 45° W–30° E. Units are W m$^{-2}$ dec$^{-1}$. Bold values are statistically significant trends at the 95 % confidence level. Note that the difference between globally adjusted $F_S$ and model-based $F_S$ in each period is the magnitude of the global adjustment (see section 3) and can also be added to other model-based $F_S$ trend estimates of that particular period.

| Averaging area | Term | DJF trend | | |
|---|---|---|---|---|
| | | 1950–84 | 1985–2019 | 1950–2019 |
| Study area | Latent heat flux | **1.3 ± 0.7** | **-1.9 ± 1.4** | -0.4 ± 0.6 |
| | Sensible heat flux | 0.2 ± 0.3 | 0.3 ± 0.3 | **0.3 ± 0.1** |
| | Radiative fluxes | **0.4 ± 0.3** | 0.1 ± 0.4 | **0.3 ± 0.1** |
| | Model-based $F_S$ | **1.9 ± 1.2** | -1.4 ± 1.9 | 0.1 ± 0.7 |
| | Globally adjusted $F_S$ | **3.2 ± 1.7** | 1.2 ± 2.2 | **1.6 ± 0.7** |
| | Inferred $F_S$ | – | 0.3 ± 1.4 | – |
| NWS box | Model-based $F_S$ | 0.8 ± 9.3 | **9.7 ± 8.0** | **4.8 ± 3.1** |
| | Inferred $F_S$ | – | **12.6 ± 9.8** | – |
| NAWH box | Model-based $F_S$ | 2.5 ± 12.4 | 5.2 ± 12.5 | 1.8 ± 4.1 |
| | Inferred $F_S$ | – | 7.0 ± 15.0 | – |
| GS box | Model-based $F_S$ | 5.9 ± 12.4 | -9.4 ± 13.2 | -0.7 ± 4.5 |
| | Inferred $F_S$ | – | -4.8 ± 12.6 | – |
| TNA box | Model-based $F_S$ | -2.9 ± 4.5 | -2.7 ± 4.2 | **-4.7 ± 1.5** |
| | Inferred $F_S$ | – | -2.3 ± 4.3 | – |
| Nordic Seas | Sensible heat flux | 0.0 ± 2.4 | 1.9 ± 2.1 | **0.9 ± 0.8** |

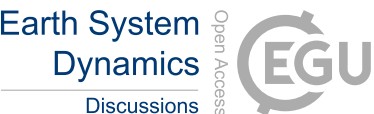

**Table 2.** Full-year trends of globally adjusted air-sea heat fluxes from ERA5 forecasts, ocean heat content tendency (OHCT), and indirectly estimated ocean heat transport (OHT) at 26° N for the period 1950–2019. $F_S$ and OHCT are spatially integrated over the ocean area between the choke point in the north (see red lines in Fig. 1) and 26° N. Trends are estimated based on full-year monthly means and 5-year means. Bold values are statistically significant trends at the 95 % confidence level.

| Method | Term | Full-year trend | |
| --- | --- | --- | --- |
| | | [W m$^{-2}$ dec$^{-1}$] | [×10$^{-2}$ PW dec$^{-1}$] |
| Monthly mean | Globally adjusted $F_S$ | **0.9 ± 0.4** | **2.3 ± 1.1** |
| | OHCT | 0.2 ± 0.9 | 0.5 ± 2.2 |
| | OHT at 26° N | -0.7 ± 0.9 | -1.8 ± 2.1 |
| 5-year mean | Globally adjusted $F_S$ | **0.9 ± 0.5** | **2.3 ± 1.3** |
| | OHCT | 0.2 ± 0.4 | 0.5 ± 1.0 |
| | OHT at 26° N | **-0.7 ± 0.6** | **-1.8 ± 1.4** |