# Peer review of "A quantitative assessment of air-sea heat flux trends from ERA5 since 1950 in the North Atlantic basin"

_Earth System Dynamics, 2023_

## Author Response (AR1)

**REVIEWER #1**

J. Mayer et al. organized a very nice study to assess the air-sea heat flux trends from ERA5 data since 1950 in the North Atlantic. Although the main focus is on the winter months, the annual changes are also investigated. The drivers of air-sea heat flux change are examined with a linearization approach. A very novel aspect is analyzing the analysis increments, which helps to show the impacts of observational system change and assess the reliability of trends. Furthermore, the air-sea flux data from ERA5 are compared with the indirect estimate based on the atmospheric energy budget approach. The ocean meridional heat transport is derived in the North Atlantic Basin. In summary, it is a very thorough analysis and a well-written manuscript. This study is also helpful for the understanding of the Earth's energy flow. I don't have any major concerns but only several clarification questions.

We would like to thank the reviewer for the insightful comments, which definitely helped to improve the manuscript. Please find below a reply to the individual comments. In addition, we revised the naming of air-sea heat fluxes as follows to avoid any confusion:

Net air-sea heat fluxes are abbreviated as $F_S$.  Net air-sea heat fluxes from ERA5 forecasts are denoted as  "model-based $F_S$" (or "model-based heat fluxes"). Indirectly estimated net air-sea heat fluxes derived from the atmospheric energy budget are denoted as "inferred $F_S$" (or "inferred heat fluxes"). Individual components of $F_S$ are called "radiative fluxes" and "turbulent heat fluxes" (or "latent heat fluxes" and "sensible heat fluxes"). These naming conventions are now consistently applied and should be clear to the reader after reading the data and methods section.

Comments:

Page-3, line-90: it is XBT, not CBT.

Thanks, we changed the text from "CBT and MBT data" to "from expendable and mechanical bathythermographs".

Page-6, line-175-180: It is not very clear how the residual has been dispersed in the Atlantic Basin, i.e. you may get a global residual, did you disperse a part of this residual (according to the area of the Atlantic Basin) or the global value?

It is the former. The global value in Wm/2 is subtracted uniformly at each grid point of the global ocean; that is, the adjustment (in Watts) in the North Atlantic is the global mean

residual weighted by the North Atlantic's area relative to the global ocean surface. To clarify this, we changed this sentence to:

"Additionally, we adjust net heat fluxes by subtracting the monthly difference between global ocean mean vertically integrated 0--2000m OHCT and $F_S$ (denoted as R) uniformly from each grid point of the global ocean, as done, e.g. by Trenberth et al. (2019)."

Page-11, line-335-338: this might be arguable: the correlation may largely come from inter-annual variation instead of trend, thus the strong correlations might not necessarily explain the trend?

Good point, we removed this sentence and added the following text:

"The skin temperature of the ice free ocean shows a statistically insignificant increase in wide areas of the tropical North Atlantic and a decrease almost everywhere north of 25°N, which coincides with the basin-wide trend pattern of the 0–300m OHC trend from IAP (not shown). Although their pattern correlation is weak (~0.2) for the ice-free ocean area due to small scale differences, the generally good match of the large-scale trend pattern in terms of sign suggests that the weakening of sensible heat fluxes is related to the ocean cooling that occurred during that time [see Hodson et al. (2014)]. For comparison, the pattern correlation for the more recent period 1985-2019, where both skin temperature and OHC increases uniformly (except for the North Atlantic Warming Hole) due to global warming, is only ~0.15 for the ice-free ocean. This indicates a generally weak spatial correlation between OHC from IAP and skin temperature from ERA5, although their large-scale trend pattern agrees well in terms of sign."

Section 4.4, the analyses related to NAO, again, the regression of NAO might largely associate with inter-annual fluctuation, so the low-frequency variation has been overlooked. It might be better to try applying a low-pass filter to the NAO index, so the decadal to multi-decadal variation can be highlighted, which is more relevant to the comparison with trends.

Thank you for this recommendation. We tested a LOWESS and Butterworth low-pass filter to smooth the NAO index. Both filters yield 70-year NAO trends of between 0.08 and 0.13 dec$^{-1}$, whereas the trend of the unfiltered timeseries is 0.11 dec$^{-1}$. Consequently, the regression on FS leads to results very similar to that shown in the manuscript. We also experimented with different parameters of the filter, but the outcome do not vary much. Thus, we would prefer to use the unfiltered NAO timeseries as its interpretation is straightforward and results are

independent of the strength of smoothing. This is now discussed in the manuscript as well. We added the following text in  section 4.4:

"To check the potential role of decadal NAO variability, we tested the impact of applying LOWESS and Butterworth low-pass filters on the NAO series before trend computation (not shown), but the impact on the trend results was small and the results are thus deemed robust."

Fig 9 (right panel): is there an error bar for both RAPID value and your estimate in black line?

We added error bars for both the observed mean transport as well as indirectly estimated mean OHT.

It would be better to put Fig. B1 into the main manuscript.

Although Fig. B1 helps to better understand the discussion about analysis increments in section 4.1, it does not contain any main results of our study. Analysis increments are used only as uncertainty estimate. Given that analysis increments and especially their decadal trends represent a quite involved diagnostic, we would prefer to leave that figure in the appendix.

**REVIEWER #2**

Review of "A quantitative assessment of air-sea heat flux trends from ERA5 since 1950 in the North Atlantic basin" by Johannes Mayer, Leopold Haimberger and Michael Mayer for Earth System Dynamics, 2023.

This paper makes a thorough investigation of air-sea heat flux trends using ERA5 forecasts. The trends in analysed variables are also studied and the impact of analysis increments is discussed. The influence of trends in the Hadley circulation, NAO and AMO are also discussed. In combination with the ocean heat content, the trends of air-sea heat fluxes are used to estimate the trend of the AMOC. On the whole, this paper is well written and the results justify the conclusions. However, there are a couple of issues that need addressing.

It is stated that "long-term changes in net surface heat fluxes over the North Atlantic Ocean are primarily driven by latent heat flux trends". These trends are -1.9 W m-2 dec-1 for 1985-2019, and the analysis increments suggest there should be an increase in the magnitude of these trends. However, the adjusted and inferred net surface heat flux trends have the opposite sign, so presumably the sensible heat fluxes are more important than the latent heat fluxes. There should be more discussion of this issue and the uncertainties.

There are a few areas where there is a lack of clarity. There is a slight confusion between net surface heat fluxes, surface heat fluxes and surface turbulent heat fluxes and it is not clear when the radiative fluxes are included. Fs is not defined. In Section 4.2 it is not always very clear which period is being discussed.

We would like to thank the reviewer for this comprehensive review and constructive comments helping to improve the manuscript. Please find below a reply to the individual comments. In addition, we revised the naming of air-sea heat fluxes as follows to avoid any confusion:

Net air-sea heat fluxes are abbreviated as $F_S$. Net air-sea heat fluxes from ERA5 forecasts are denoted as "model-based $F_S$" (or "model-based heat fluxes"). Indirectly estimated net air-sea heat fluxes derived from the atmospheric energy budget are denoted as "inferred $F_S$" (or "inferred heat fluxes"). Individual components of $F_S$ are called "radiative fluxes" and "turbulent heat fluxes" (or "latent heat fluxes" and "sensible heat fluxes"). These naming conventions are now consistently applied and should be clear to the reader after reading the data and methods section.

pg 2, ln 53: Can you comment on why these fluxes are only available for this limited period?

To explain the limited availability, we added the following text: "[due to the limited availability of observationally constrained TOA fluxes, see Liu et al. (2020) and Mayer et al. (2022)],".

pg 3, ln 72-73: Which of these variables are on the lowest model level? Is surface pressure used too?

To make this clear, we changed the text from: "We also use single-level atmospheric moisture, temperature, pressure, and 10 metre wind fields as …" to

"We also employ temperature and moisture in the lowest model level as well as surface pressure, 10 metre wind, near-surface atmospheric density, and skin temperature fields as …".

pg 3, ln 75; pg4, ln 99: What do you mean by "input parameters" and "input variables"?

We changed "input parameters" in line 75 to "input variables" and added "(moisture, temperature, and wind speed; trends in air density are in general not discussed as they are negligible small)".  To employ a consistent naming throughout the manuscript, we now use "variables"  everywhere, instead of  "parameters".

pg 3, ln 83: The link is broken.

It's a direct link to the official ftp server, it might need a confirmation to proceed. We replaced it with the link to the official website:
http://www.ocean.iap.ac.cn/pages/dataService/dataService.html?navAnchor=dataService

pg 5, ln 127: "12 hourly" or "12 hour"? Are analysis increments calculated just with 12 hour forecasts ie at 06/18 UTC?

Thanks for the comment, this sentence could indeed be misunderstood. Yes, analysis increments are calculated just with 12-hour forecast at 06 and 18 UTC to show their largest possible impact (difference between forecast and analysis might be smaller for shorter forecast lead times). We changed the sentence accordingly: "… are calculated at 06 and 18 UTC from the difference between analyzed state quantities and 12-hour forecasts valid at those times".

pg 5, ln 127-129: "The impact of increments .." - more explanation required.

We changed the sentence from: "The impact of increments on flux trends is then estimated ..." to "Uncertainties in model-based $F_S$ trends caused by analysis increments are thus estimated ... " because the difference in trends between analyzed and forecast fluxes gives us a rough uncertainty estimate. The exact impact of the increments on trends is actually unknown.

We also added the following sentence: "During data assimilation, changes in the observing system can introduce analysis increments that may add or remove temperature or humidity from the model resulting in unrealistic trends (Bengtsson et al., 2004; Trenberth et al., 2011; Hersbach et al., 2020; Mayer et al., 2021)."

pg 6, ln 170, equ 5: The units of OHT would be Wm-2, or are the 2nd and 3rd terms on the RHS area integrals, not means? Table 2 gives the trends in OHT in units of W m-2 dec-1 and PW dec-1.

To clarify this, we changed this sentence to: "..., and the second term on the right side is the difference between net surface heat flux ($F_S$; from ERA5 forecasts) and temporal ocean heat content tendency (OHCT; from 0–2000m IAP data) integrated over the ocean area between $\varphi_C$ and $\varphi$ (the Mediterranean Sea is excluded; units are Watts)."

We also added the following text in the caption of Table 2: "The left column contains area-averaged values (relative to the area between the northern choke point and 26° N) given in W $m^{-2}$ $dec^{-1}$. The values in the right column represent the area-integrated contribution to the ocean heat budget given in PW $dec^{-1}$."

pg 6, ln 170, equ 5: Fs is not defined.

$F_S$ is now defined in the methods section, see response to previous comment.

pg 8, ln 216-217: Prominent positive trends in "the Norwegian and Labrador seas" is not very exact. They extend beyond the Norwegian sea and the Labrador sea has already been mentioned for negative trends.

That's true, we changed the text from: "Positive trends (...) are prominent in the Norwegian and Labrador Sea, but also ..." to "Positive trends (...) are prominent in the Nordic Seas and in the eastern part of the Labrador Sea, but also ..." and added "the western part of the Labrador Sea" when negative trends are mentioned. Now it should be clear to the reader which regions are meant.

pg 8, ln 236/238: It might be worth putting these numbers in a table.

Thank you for your suggestion. After careful consideration, we believe that presenting them in a table may disrupt the overall flow and readability of the paragraph.

pg 9, ln 258-259: Are the reduced trends solely in the Irminger Sea or in it's vicinity?

Good question, the explanation in the manuscript is not quite accurate. The decreasing trends are actually located south of the Irminger Sea. To clarify this, we changed the text from: "... decreasing trend in the Irminger Sea, which ..." to

"... decreasing trend to the south of the Irminger Sea (around 20-30° W, 55° N), which ..."

pg 9, ln 260-261: Figs. 3e and f should be mentioned here.

Done.

pg 9, ln 273-274: A more exact description of the location of the Gulf stream region, among regions of strong sea-ice retreat is required.

We changed the text from "Among the regions of strong sea ice retreat, peak positive trends in surface humidity and temperature can be found along the Gulf Stream region ( ...)." to

"Among the regions of strong sea ice retreat in the Labrador and Nordic Seas, peak positive trends in surface humidity and temperature can be found along the main current of the Gulf Stream (35-45° N, 80-45° W; ...)." The region of sea ice retreat is defined in section 4.1.

pg 10, ln 287, point 6: You might expect a similar mechanism for humidity - can you comment?

Good point. There might be several reasons why this effect is not visible for latent heat fluxes as well, but this would need further investigations that are beyond the scope of this work. One explanation could be that the surface latent heat fluxes in regions of strong sea ice retreat are balanced (or probably overcompensated) by the atmospheric moisture transport (advection) so that the signal of the surface saturation humidity trend (Fig. 3c) is barely visible in changes of the model-level humidity (Fig. 3a). Consequently, atmospheric moisture further south would be almost unaffected by the stronger latent heat fluxes over regions of

strong sea ice retreat. Furthermore, trends in the near-surface atmospheric moisture seem to be unaffected by the exact location of sea-ice retreat (there is only a very weak signal at 60-70°N, 60°W, associated with the sea-ice retreat, see Fig. 3a). Precipitation could also play an important role. Since this is highly speculative, we prefer to only add "This effect cannot be observed for latent heat fluxes and requires further investigation that is beyond the scope of this study.".

pg 10, ln 299-301: Either a reference or more explanation is required to justify the link between increased low level cloud and OLR, and an intensified Hadley Cell.

We added the references Chen et al 2002 (doi:10.1126/science.1065835) and Mathew et al. 2019 (10.1016/j.jhydrol.2018.12.047). Chen et al. discusses the increase in thermal radiation in the tropics, Mathew et al shows trends of low, mid, and high level cloud fraction (see Fig. 8 therein).

pg 11, ln 315: What do you mean by "artificially"? The data assimilation blends the background forecast with observations, to obtain the best fit to both, taking account of the known errors of both.

Good point. The word "artificially" might not be necessary, because the removal of moisture by analysis increments is always artificial, and not a real signal. We removed it.

pg 11, ln 324-325: "..long-term changes in net surface heat fluxes over the North Atlantic Ocean are primarily driven by latent heat flux trends (Table 1).." - is this the case? The LHF trends are about -1.9 W m-2 dec-1 (Table 1) and the analysis increments increase the magnitude of these trends by 1-2 W m-2 dec-1. Only the adjusted and inferred Fs trends are positive. So, surely the SHF trends are important too.

We revised the last two paragraphs of section 4.1 to make the consequences of analysis increments clear.  It should be noted that the discrepancy between model-based and inferred fluxes cannot solely be explained by the increments shown in this manuscript (i.e., as difference between forecasts and analyses). Changes in the observing system may have an influence on both forecasts and analyses, albeit the exact impact on the trends is unclear. Analysis increments and the difference in fluxes computed with forecasts and analyses can only be used as rough uncertainty estimate as mentioned in the manuscript. In other words, the computed moisture (and temperature) difference between forecast and analysis as shown in appendix B is just an indicator of where changes in the observing system may have an impact on the trends, although the exact magnitude of this impact is unclear. Consequently, the temporal discontinuities in the late 1990s and early 2000s do not vanish

when using analyzed fields instead of forecasts to compute surface fluxes based on bulk formulas. It requires the application of the global adjustment. With that being said, the careful statement made in these lines holds true.

To further clarify this, we added the following text in section 4.1: "It should be highlighted that analysis increments computed in this way cannot be used to fully remove the temporal inconsistencies in model-based $F_S$ as their exact impact on both forecasts and analyzed state quantities is unknown. That is, the difference between adjusted and unadjusted model-based fluxes (magnitude of the global adjustment) is not equivalent to the difference between fluxes computed with forecasts and analyzed state quantities."

pg 11, ln 333-334: Can you explain more what you mean by "strongest weakening"?

To clarify this, we changed the sentence to: " ..., the region with the most prominent negative trends after the 1980s (...)."

pg 11, ln 334-339: It's not clear which period(s) is(are) being discussed here. Is the earlier or later period being discussed or both? More clarity is required.

We revised this paragraph and explicitly mentioned (in the 2$^{nd}$ and 3$^{rd}$ sentence)  that the trends before 1985 are discussed.

pg 12, ln 344-349: This paragraph is unclear. Which period is this for? What is "CC-related decrease"? "Negative trends" of what?

This section/paragraph describes trends for the period 1950-1984. We revised the whole paragraph and the paragraph before to clarify these points. CC was the abbreviation for Clausius Clapeyron and is now written out.

pg 12, ln 359: What is the relevance of Table 1?

We removed the reference to table 1 in this line and mentioned it in the first sentence of this section instead. Table 1 provides exact values for the trends shown in Fig. 7.

pg 12, ln 365-373: Are these lines discussing the anomalies or partial trends, or both, in Fig. 7?

It's about the model-based F_S trends. We revised this paragraph, especially the first sentence to clarify this.

pg 13, ln 401; (pg 15, ln 475): "heating in the western North Atlantic" - do you mean where the positive trends are, maximised over the Gulf Stream? The western North Atlantic is a big area, so you need to be more specific.

Exactly, "heating in the western North Atlantic" is related to the positive trend over the Gulf Stream. To clarify this, we added coordinates and referred to the positive trends in Fig. C1.

pg 13, ln 402: "Fig. 8a" or should it be "Fig. C1a"?

We also added a reference to Fig. C1a, see answer to previous comment.

pg 15, ln 470: "Fig. 8a" or "Figs. 2e and 6a"?

True, it should actually be Fig. 2e. Corrected.

pg 16, ln 493: Do you mean that there would be stronger negative trends from analyses?

Exactly. We added "(more negative)" to clarify this.

pg 16, ln 505: "~3" or "~2"? Explain a bit more.

The trend based on analyzed fields is -4 W m-2 dec-1, which is stronger (more negative) than the trend based on forecasts by 1.3 W m-2 dec-1, which is a factor of 3 (more precisely 3.2) smaller than the  4.2 W m-2 dec-1 statistical uncertainty given in table 1. This example shows how analysis increments can be used as rough estimate for trend uncertainties (as mentioned before in the manuscript).

pg 16, ln 509-510: "..we find that the global correction removes much of the effect of spurious air-sea flux trend on our inferred estimate of OHT" Can you discuss the justification for this statement a bit more?

We added the following sentence to justify this: "that is, temporal inconsistencies in basin-wide averages of model-based $F_S$ (and thus also in the inferred OHT) are almost completely eliminated by the global correction."

pg 17, ln 512-513: "Air-sea heat flux trends from ERA5 forecasts in the North Atlantic basin seem reliable in terms of sign and spatial structure," - this requires more discussion given that LHF dominate but have a negative trend, like Fs. Then the analysis increments increase

the magnitude of the negative LHF trends. Only the globally adjusted and inferred FS trends are positive.

See reply to comment about pg 11, ln 324-325.

We also added "... in terms of sign (on sub-basin scale, see Fig. 7) and spatial structure ..." to make clear that the sign of model-based flux trends is reliable on smaller spatial scales (as mentioned in the abstract).

TYPOS
pg 1, ln 7: "..sub-basin scales..".
pg 1, ln 9: "..modes of climate variability..".
pg 1, ln 9-10: "..patterns found..".
pg 1, ln 10: "..Irminger and Labrador Seas..".
pg 1, ln 21: "..the Azores High..".
pg 2, ln 31: "..the atmosphere..".
pg 2, ln 42: "..preliminary back-extension..".
pg 2, ln 59: "..the NAO..".

Thanks – all corrected.

pg 3, ln 69: ERA5 provides twice daily 12 hour forecasts in the CDS and 18 hour forecasts in MARS, all with hourly output.

We are referring to the data available in the CDS as they are accessible to everyone. We revised this sentence to highlight the 1-hourly resolution: "ERA5 provides hourly estimates of a variety of meteorological variables as analyzed state quantities as well as 12-hourly twice-daily forecasts on a Gaussian grid ..."

pg 3, ln 75: "..which allows an estimation of the role of..".
pg 3, ln 76: "..a regular..".

Thanks – both points corrected.

pg 3, ln 89-90: Are CBT and MBT defined?

We changed the text from "CBT and MBT data" to "from expendable and mechanical bathythermographs".

pg 3, ln 92: "..the northern choke point.."?
pg 4, ln 113/115: "..the lowest model level..".
pg 6, ln 176: "..arrays..".
Thanks – all corrected.

pg 7, ln 186: rho was used as density previously eg equations 1, 2, 3.

Good point, gamma  is now used for the autocorrelation function.

pg 7, ln 187: "..these conditions..".
pg 7, ln 203: "..an immense..".
pg 8, ln 212: "..in Baffin Bay and the Labrador Sea..".
pg 8, ln 214: "..or regions of..".
pg 8, ln 220: "..peak values..".
pg 8, ln 225: "The main contributors to the Fs trends..".
pg 8, ln 228: "..the inferred..".
pg 9, ln 249: "..the lowest model level..".
pg 10, ln 287: "..downwind of areas of..".
pg 10, ln 307: "..as a rough estimate..".

Thanks – all corrected.

pg 11, ln 309-310: It might be worth specifying that the trend is negative.

We revised this sentence,  it now should be clear that the trend is negative.

pg 11, ln 313: "..from analyses, for the North Atlantic zonal mean, are.."
pg 12, ln 350: "..as a possible source..".
Thanks – all corrected.

pg 12, ln 361: Where is the wind direction information in Fig. 7?

We added a reference to Fig. 4a.

pg 12, ln 361: Maybe "dominate" would be better than "overcompensate".
pg 13, ln 378: "..opposite directions..".

pg 13, ln 380: "In addition, the partial trends demonstrate .." -is this what you mean?

pg 14, ln 437: "The main contributors..".

pg 15, ln 453: "..allowed us to quantitatively..".

pg 15, ln 455: "..discussed their reliability..".

pg 16, ln 491: "..at the lowest model..".

Thanks – all corrected.

FIGURES AND TABLES

Figure All: The labels are very faint. Difficult to see statistical significance shading when plot is small.

We revised the labels and the shading of all figures.

Figure 2 caption: Are the net surface heat fluxes in a and b, Fs?

Yes, to clarify this we changed the text to "model-based $F_s$" and "inferred $F_s$" (see comment above).

Figure 2 caption: "Panels.."; "..flux components.."; "..panels.."; "..computed from DJF..".

Corrected.

Figure 3: Labels are very faint. Difficult to see statistical significance shading when plot is small.

We revised this figure.

Figure 7: There are two, different, columns for ws in the partial trends.

Yes, because the wind speed has to be regressed on both latent heat and sensible heat fluxes, see Eq. (3) and explanation therefor. As the trends in latent and sensible heat fluxes are different, the regression (i.e., the 'sensitivity') yields different partial trends for ws.

Figure 7 caption: For which period are the climatologies in the middle panel?

They are based on 1985-2019 to get consistent anomalies between inferred and model-based fluxes. The caption stated: "Anomalies are computed w.r.t. 1985–2019". To clarify this, we changed this sentence to "Climatologies and anomalies are computed w.r.t. 1985–2019".

Figure 8 caption: Are the surface heat fluxes, the net fluxes (Fs?)?

Good point. We changed "model-based surface heat fluxes from ERA5 forecasts" to "model-based $F_S$" (see comment above).

Figure 9: The shading is not easy to interpret.

We changed the shading in this figure for better interpretation.

Figure B1 caption: "..at the lowest..".

Corrected.

Table 2: Are the 2 columns of results required? If so, they should be mentioned in the caption.

The two columns for different units are now mentioned in the caption (see also reply to ln 170).